# Synthesis of Silver Modified Bioactive Glassy Materials with Antibacterial Properties via Facile and Low-Temperature Route

**DOI:** 10.3390/ma13225115

**Published:** 2020-11-13

**Authors:** Isabel Gonzalo-Juan, Fangtong Xie, Malin Becker, Dilshat U. Tulyaganov, Emanuel Ionescu, Stefan Lauterbach, Francesca De Angelis Rigotti, Andreas Fischer, Ralf Riedel

**Affiliations:** 1Institut für Materialwissenschaft, Technische Universität Darmstadt, Otto-Berndt-Straße 3, D-64287 Darmstadt, Germany; xiefangtong@gmail.com (F.X.); m.l.becker@utwente.nl (M.B.); tulyaganovdilshat@gmail.com (D.U.T.); emanuel.ionescu@tu-darmstadt.de (E.I.); ralf.riedel@tu-darmstadt.de (R.R.); 2Department of Natural–Mathematical Sciences, Turin Polytechnic University in Tashkent, 17, Small Ring, Tashkent 100095, Uzbekistan; 3Institut für Angewandte Geowissenschaften, Technische Universität Darmstadt, Schnittspahnstrasse 9, D-64287 Darmstadt, Germany; stefan.lauterbach@geo.tu-darmstadt.de; 4Division of Vascular Signaling and Cancer (A270), German Cancer Research Center (DKFZ), Im Neuenheimer Feld 280, 69120 Heidelberg, Germany; f.deangelisrigotti@dkfz-heidelberg.de (F.D.A.R.); andreas.fischer@medma.uni-heidelberg.de (A.F.); 5Department of Medicine I and Clinical Chemistry, University Hospital of Heidelberg, 69120 Heidelberg, Germany

**Keywords:** bioactive glass, antibacterial, silver, nanocomposites, *E. coli*, ion release

## Abstract

There is an increasing clinical need to develop novel biomaterials that combine regenerative and biocidal properties. In this work, we present the preparation of silver/silica-based glassy bioactive (ABG) compositions via a facile, fast (20 h), and low temperature (80 °C) approach and their characterization. The fabrication process included the synthesis of the bioactive glass (BG) particles followed by the surface modification of the bioactive glass with silver nanoparticles. The microstructural features of ABG samples before and after exposure to simulated body fluid (SBF), as well as their ion release behavior during SBF test were evaluated using infrared spectrometry (FTIR), ultraviolet-visible (UV-Vis) spectroscopy, X-ray diffraction (XRD), electron microscopies (TEM and SEM) and optical emission spectroscopy (OES). The antibacterial properties of the experimental compositions were tested against *Escherichia coli* (*E. coli*). The results indicated that the prepared ABG materials possess antibacterial activity against *E. coli*, which is directly correlated with the glass surface modification.

## 1. Introduction

Among the different synthetic biomaterials aimed for regenerating bone defects, bioactive glasses attract attention due to their controllable chemical dissolution and bonding to the target tissue facilitating its healing in a relatively short period of time [1,2,3].

One of the main issues connected to orthopedical surgery are bacterial infections that might lead to implant failure, devastating health complications for the patient, and high treatment costs [4]. Van de Belt et al. [5] reported that, when orthopedic implants are in contact with blood, plasma proteins can form a conditioning film on the surface of the implant where microorganisms can easily adhere to. Current efforts are focused on the modification of the surface of bioactive materials with inorganic antibacterial elements to reduce the use of antibiotics and the associated drug resistance issues. Antibacterial properties can be provided to glasses typically by ion exchange either in aqueous solutions or molten salts. Among various inorganic antibacterial agents, silver has been demonstrated as a suitable candidate for the ion exchange approach due to its great ability to enter the silicate glass structure. Moreover, it exhibits a broad-spectrum bactericidal behavior at low concentrations without causing resistant bacteria [6,7,8,9], while keeping the bulk structure and main properties of the material mostly unaltered [10,11,12,13,14]. Silver ions can be easily exchanged with the Na^+^ ions of the bioactive glass due to their similarities in ionic radius and valence. Silver ions can be deposited on the surface or penetrate into the glass network, strongly depending on the reaction conditions used for preparing the silver modified materials. Verne et al. [15] demonstrated that silver ions introduced into a bioactive glass (SiO_2_-CaO-Na_2_O) network and/or deposited on the surface by ion exchange in aqueous solution provided better control of the amount of incorporated silver as compared to the Ag-modified materials prepared by the molten salt approach, which enables avoiding cytotoxic effects [15]. The main issue related to ion exchange in aqueous solutions relates to the associated long reaction times, lasting from days to months [11,15]. In this regard, sonochemistry has been demonstrated as a suitable approach to reduce mainly bulk high temperatures, high pressures, and long reaction times involved in some methods [16]. The production of silver nanoparticles from aqueous silver nitrate solutions sonochemically was investigated by Mănoiu and Aloman [17]. The proposed method consisted of applying a strong flow of ultrasonic energy [17,18] to silver nitrate solutions with concentrations ranging from 0.1 M to 0.001 M at 30 °C for 1 h. These conditions led the fluid to cavitate (i.e., formation, growth and implosive collapse of bubbles) [19] providing quasi-spherical silver nanoparticles of about 7 nm average particle size. He et al. [19] reported on the formation of Ag nanoparticles by an ultrasonic approach in acidic, neutral and alkaline aqueous media without using any reductant and surfactant reagent. They proposed that upon sonication of aqueous silver nitrate solution, water molecules decompose to hydrogen and hydroxyl radicals (H∙ and ∙OH, respectively). Finally, H∙radicals supply electrons to Ag^+^ which is reduced to metallic silver [19].

In the present work, silver from a AgNO_3_ solution has been incorporated into the surface of a bioactive glass which contains magnesium, fluorine and features significantly lower sodium content as compared to that of the well-known 45S5 Bioglass^®^ [20,21]. The combination of the ion exchange approach with the ultrasonic treatment has been shown to represent a facile, time-saving method to modify the surface of a bioactive glass with Ag. The structural characterization of the Ag-modified glass samples was performed using X-ray diffraction, UV-Vis spectroscopy, TEM and SEM microscopy before and after exposure of the samples to simulated body fluid for several time spans. The antibacterial activity of silver-containing BG has been assessed using *E. coli* strain.

## 2. Materials and Methods

### 2.1. Preparation of Silver-Modified Glass Materials

The bioactive glass (BG) investigated in this work has the following composition: 4.33Na_2_O-30.30CaO–12.99MgO–45.45SiO_2_–2.60P_2_O_5_–4.33CaF_2_ (mol%). It was selected due to beneficial results demonstrated during its application in regenerative biomedicine [20,21,22]. Fine particles with sizes below 32 μm were prepared following the approach reported elsewhere [20,21,22]. The calculated network connectivity of the BG glass investigated in the present study is about 2.25, and thereby can be considered bioactive according to Hill et al. [23].

Silver-modified (ABG) glassy materials were prepared by a sonochemical method using fine powders of BG and 0.035 M (0.60 wt.%), 0.077 M (1.25 wt.%), 0.150 M (2.50 wt.%) and 0.220 M (3.70 wt.%) silver nitrate (ACS reagent, ≥99.0%, Sigma Aldrich, St. Louis, MO, USA) solutions. The experimental silver modified ABG materials were further denoted as ABG1, ABG2, ABG3 and ABG4, respectively. The solid load of BG powder in the AgNO_3_ solutions was kept as 10 mg/mL (i.e., 1 g of BG was suspended in 100 mL of solution). To avoid photodegradation, the laboratory ware accommodating silver nitrate solutions and ABG glassy materials was covered by aluminum foil. The procedure for ABG glassy materials preparation is summarized in Scheme 1 and may be described as follows: BG powder was stirred in AgNO_3_ solutions for 20 min and then a flask with the obtained suspension was placed in the ultrasonic ice-water bath at 35 kHz for 30 s. This procedure of sonochemical treatment was repeated 3 times. Then the sonicated suspensions were heat treated at 80 °C for 20 h in air. After centrifugation, the solid was washed with deionized water and dried at 37 °C for 24 h.

### 2.2. Physicochemical Characterization

X-ray diffraction (XRD) patterns of the as-prepared samples were taken with a STADI P X-ray diffractometer (STOE&Cie GmbH, Darmstadt, Germany) using Mo Kα radiation (λ = 0.7093 Å), 45 kV voltage and 30 mA current. The spectra were performed from 5° to 40° in the 2θ range. For performing the Fourier transform infrared (FTIR) spectroscopy measurement the samples were mixed with KBr (mass ratio: sample:KBr = 1:150) and pressed into a pellet. The spectra were recorded in a 670-IR spectrometer (Varian Inc., Santa Clara, CA, USA) from 400 cm^−1^ to 4000 cm^−1^. A JEM 2100F transmission electron microscope (JEOL Ltd., Akishima, Tokyo, Japan) was used for generating the high-resolution transmission electron microscopic (HRTEM) images of the ABG samples. Scanning electron microscopy (SEM) analysis was executed on a XL30 FEG (Philips, Chatsworth, CA, USA) under 10–15 kV acceleration voltage. The Ultraviolet-visible diffuse reflectance (UV-vis DR) experiments were recorded at room temperature on a Lambda 900 UV/VIS/NIR Spectrometer (Perkin Elmer, Waltham, MA, USA).

### 2.3. In Vitro Acellular Mineralization Tests

The capability of the silver modified glasses of inducing the formation of calcium phosphate phases onto the glass surfaces, was assessed by soaking 75 mg of the as-prepared samples in 50 mL of a protein- and cell-free simulated body fluid (SBF) at 37 °C. The SBF solution was prepared according to the procedure described by Kokubo et al. [24]. The experiment (static) was performed in airtight polyethylene flasks, at 37 °C for: 1 day (1D), 2 days (2D), 3 days (3D), 7 days (7D), 12 days (12D), 14 days (14D), 21 days (21D) and 28 days (28D). At the end of each time period, an aliquot was removed from the sample and the solids were separated from the liquid by filtration. After rinsing the powders with deionized water and acetone, they were characterized by means of FTIR, XRD and SEM. The pHs of the filtered solutions were immediately measured and their Si, Na, Ca, Mg, P and Ag concentration was determined by using an inductively coupled plasma-optical emission spectrophotometer (Aligent 720 ICP-OES; Aligent, Santa Clara, CA, USA). Additionally, ABG4-chitosan composites were prepared and the same SBF test described before for the silver modified glasses was performed. For the preparation of 1 g of ABG4-chitosan composite, 20 mg of chitosan (medium molecular weight, Sigma Aldrich, Darmstadt, Germany) were dissolved in aqueous acetic acid. Subsequently, 8.6 mg of ABG4 powder suspended in deionized water were added dropwise, under continuous stirring, to the chitosan solution until a homogeneous composite containing 2 wt.% chitosan and 0.86 wt.% ABG4 in 0.1 M acetic acid was obtained.

### 2.4. Antibacterial Activity

Experiments of bactericidal activity have been performed on TOP10 chemically competent *Escherichia coli* (*E. coli*, Thermo Fisher Scientific, Waltham, MA, USA). Bacteria were previously transformed and carried a plasmid containing ampicillin-resistance cassette. Therefore, bacteria were resistant to ampicillin, facilitating the handling in non-sterile conditions. By using ampicillin in both culture media and plates, contamination by environmental bacteria is prevented. *E. coli* were grown at 37 °C in Luria-Bertani medium (LB medium, Sigma, St. Louis, MO, USA) in presence of ampicillin (100 μg/mL, Sigma, St. Louis, MO, USA) and constantly shaken at 180 RPM.

Two experimental compositions namely ABG3 and ABG4 were sterilized by autoclaving and subsequently added to sterile LB medium (with 100 μg/mL of ampicillin). For comparative purposes the unmodified bioactive glass BG was also tested. Two experiments were performed to investigate the antibacterial activity of the samples.

#### 2.4.1. Analysis of Bacterial Growth Inhibition

Ampicilin-resistant *E. coli* (~10^7^ colony forming units, CFU) were plated on LB broth agar plates (Sigma, St. Louis, MO, USA) with ampicillin and four areas of the plates were treated with 10 μL of BG and silver-modified BG glass particulates (200 mg/mL). Upon incubation at 37 °C for 24 h, plates were analyzed for signs of bacterial growth inhibition. Bacteria grow until covering all the surface of the plate with a semitransparent/white layer. When an area is treated with a growth inhibition factor, it remains transparent, due to the absence of bacterial growth.

#### 2.4.2. Analysis of Bactericidal Action over Time

Experiment was performed in 96-well plates with U-bottom shape. Ampicilin-resistant *E. coli* were cultured in a total volume of 200 μL, which consisted of 100 μL of bacteria inoculum and 100 μL of glass particulates solution (20 mg/mL). Therefore, the final working concentration of glass particulates was 10 mg/mL. Amount of bacteria was evaluated by CFU at the moment of the inoculum, and after 1 h, 2 h and 4 h of incubation at 37 °C with glass particulates. To calculate the CFU, bacteria were vortexed to have a solution of single bacteria and serial dilutions (1:10 for 6 times) were plated on LB broth agar plates (with 100 μg/mL of ampicillin) for 24 h at 37 °C. After incubation, the plates in which colonies were overlapping (too low dilution) and the ones in which no colonies appeared (too high dilution) were discarded. On the remaining plates, number of colonies was counted and multiplied for the appropriate dilution factor (depending on the serial dilution) to obtain number of bacteria (CFU).

The statistical differences between experimental conditions were calculated according to the Student’s *t*-test of Graph Pad software (Prism 8 version). The samples were considered significantly different when the *p* value was less than 0.05.

## 3. Results and Discussion

### 3.1. Chemical and Microstructural Characterization of Silver Modifed Glasses (ABG)

The XRD patterns of the samples investigated within this study are shown in Figure 1a. The XRD patterns indicate that BG is X-ray amorphous, where a broad hump centered at approximately 13° is assigned to the amorphous silica network. The prepared ABG glassy materials are of amorphous nature as well, thus, the reflections with low intensity observed in the XRD patterns may be ascribed to crystalline silver (I, II, and III) oxide phases (Ag_2_O, Ag_3_O_4_, and Ag_3_O) [25,26,27] and elemental silver (Ag) [28]. The presence of elemental Ag was also corroborated by UV-Vis spectroscopy. Figure 1b shows the UV-Vis spectra of the ABG glassy samples prepared using solutions with different concentrations of silver nitrate. Silver nanoparticles (NPs) interact with light strongly due to their known surface plasmon resonance (SPR) [29]. Typically, the SPR peak of the Ag NPs is located between 390 nm to 476 nm. The absorption bands of the ABG samples are located in the visible range (from ca. 350 nm to 550 nm) with the plasmon peak centered at 392 nm. The detection of the SPR peak, which was not present in the parent glass (BG), confirms the presence of the elemental silver in all as-prepared ABG samples [19].

The HRTEM image (Figure 2a) reveals the presence of silver nanospheres with an approximate mean diameter of 10 nm. These nanospheres tend to aggregate on the surface of the glass forming silver clusters of different sizes with quasi-spherical shapes (Figure 2b). Some case studies related to the Ag incorporation into the surface of bioglasses reported on the presence of needle-shaped Ag clusters, which were discussed within the context of possibly damaging the cell walls of the bacteria and consequently inhibiting their activity [30]. However, as the silver clusters in the present study are spherical, one may expect that physical damages of the cell walls are rather unlikely.

One question that may arise at this point is regarding the origin of elemental silver. Since neither natural nor chemical reductants have been directly used during the preparation of the ABG samples, most likely, silver nanoparticles were formed during the sonochemical treatment. It is known that the ultrasonic irradiation of the H_2_O generates highly reactive species (mainly H∙ and ∙OH) which might be responsible for reducing silver ions to elemental silver [19]. On one hand, according to He et al. [19], the reductive rate of Ag^+^ in pure water via ultrasonic irradiation is very low; however, it may be enhanced by increasing the concentration of OH^−^ in the fluid. It is known that when silicate glasses, like the bioactive glass investigated in the present work, are in contact with water ion exchange reactions between modifiers (such as Na^+^, Ca^2+^ and Mg^2+^) and H^+^ from the surrounding fluid take place at the glass/liquid interface leading to a local (near the surface) increase of the concentration of OH^−^ [31]. AgNO_3_ in alkaline media forms Ag(OH)_x_ species in equilibrium with Ag_2_O [32,33], whose presence in the silver modified glasses has been shown by X-ray diffraction (Figure 1a). Then, upon performing ultrasonic conditioning Ag_2_O crystalline phases are reduced to metallic silver [19]. Contemporarily, that does not exclude the fact that along the procedures involved silver ions may be exchanged with Na^+^ ions in the upper atomic layers of the BG, as demonstrated by Verne et al. [15].

The mid infrared spectra of the investigated samples, prepared with increasing silver content from 0 to 3.7 wt.% (Figure 3a), exhibit transmittance bands attributed to characteristic vibrations of SiO_4_ tetrahedron units with different number of bridging oxygen (BO) atoms. The band located at high wavenumbers (from 1200 cm^−1^ to 850 cm^−1^) is ascribed to the Si–O asymmetric stretching mode of the non-bridging oxygens (NBOs) [34,35]. It shows that the network mainly features Q^2^ species (SiO_4_ sites with 2 BO and 2 NBOs) along with Q^3^ groups (SiO_4_ sites with 3 BOs and one NBO) [22,36]. The bands located at lower wavenumbers are ascribed to bending vibrations of the same units (bands between 730 and 800 cm^−1^) and to rocking motion of Si-O-Si units (bands between 400 and 550 cm^−1^) [22]. The structure of all silver-modified glasses investigated is similar to that of the parent glass BG, which indicates that the silver treatment performed within this work does not significantly alter the glass network.

### 3.2. In Vitro Acellular Mineralization Assessments of the Silver Modified Glasses

The formation of apatite upon exposure the glasses in SBF as well as the bacteria growth are strongly pH dependent [37]. Thus, pH evolution for BG and for ABG materials in SBF at 37 °C for different periods of time was monitored. The pH of the SBF solution was fixed as 7.4 ± 0.1. As shown in Figure 4, compared to the parent BG, in all ABG glassy materials the pH demonstrated much faster increase, while this tendency was less pronounced after 1 day of immersion. The maximum pH values recorded for the silver modified samples were 8.0 ± 0.1 after 72 h of immersion (Figure 4). The observed increase about 0.6 pH units is relatively low in contrast with the increase observed for the benchmark 45S5 which rises the pH of the SBF solution above 1 pH units [24]. The lower increment observed might be associated with the fluoride ions that are part of the BG used within this work. While the cations such as Na^+^, Ca^2+^ and Mg^2+^ from the glass surface are leached out from the glass to the solution and replaced by H^+^, F^−^ ions are exchanged with OH^−^ ions, thus depleting the concentration of hydroxide ions in the solution and buffering the effect of alkali/ alkaline earth ions [23]. In spite of that, one can notice that the pH values of the silver modified glasses are always beyond the values gathered for the parent glass. This fact might be associated with the release of silver ions during the first hours of immersion according with the evolution of silver concentration in the fluid over the time (Figure 5f) which will be explained in detail below: silver ions from the surface are exchanged with hydrogen cations from the solution, thus increasing the concentration of hydroxide anions in the solution. Conversely, the parent BG exhibited lower pH values with a plateau after the first hours of immersion until 2 days of immersion, followed by further decline.

Figure 3 illustrates mid infrared spectra obtained for the investigated materials upon exposure in SBF for 0 days (0D, Figure 3a), 1 day (1D, Figure 3b), 3 days (3D, Figure 3c), and 7 days (7D, Figure 3d). The spectra of the soaked samples clearly show that the band attributed to Q^2^ (SiO_4_ sites with 2 BO and 2 NBOs) disappeared, while the broad band ascribed to Si-O-Si rocking motions gets narrower. Bands related to the Si–O asymmetric stretching mode of the NBO (namely, Q^2^, Q^1^, and Q^0^) are detected only in the silver modified glasses after exposure in SBF (Figure 3b–d). Moreover, a single peak at ~560 cm^−1^, ascribed to P–O bending vibrations, can be observed in the spectra of both silver-modified and silver-free glasses from the beginning of the test. Since only a broad single peak is observed it may suggest the presence of amorphous, rather than crystalline, calcium phosphate phases [38].

The evolution of the ion concentration during the SBF test was assessed by optical emission spectroscopy. The leaching profiles of Si, Na, Ca, Mg, P and Ag are shown in Figure 5. It can be observed that the silicon, sodium, calcium, and magnesium concentrations evolve similarly in both ABG materials and silver-free parent glass during the first week of immersion. However, during the second week of exposure in SBF, the concentration of Si, Na, Ca and Mg increased in the ABG compositions while it was constant (e.g., Mg) or even decreased (e.g., Si, Na, and Ca) in the parent glass.

It is well documented that the removal of phosphorous from SBF solution is attributed to the precipitation of phosphate species on the glass surface. It can be observed that while the parent glass demonstrates a smooth decrease in phosphorus concentration, the ABG samples show more abrupt profiles. Importantly, the Ag leaching profiles in ABG samples (Figure 5f) suggest abrupt growth in Ag ion concentration during the first hours till 1 day of immersion with steady increase in silver concentration along all 12 days of immersions (with the exception revealed for 7 days of immersion in ABG4).

The mineralization of bioactive glasses is typically investigated by the observation of amorphous and crystalline calcium phosphate developed on the glass surface upon immersion in SBF solution. Figure 6 shows XRD patterns of sample ABG4 after immersion in SBF solution for different periods of time that revealed characteristic diffraction peaks of AgCl (ID #: 00-031-1238) [15]. As stated in the literature [15], during the SBF test, ABG4 releases Ag^+^ ions which speedily react with the chlorides from the SBF solution leading to the precipitation of insoluble AgCl crystals on the surface of the glass. Moreover, the AgCl crystals on the surface may prevent the direct contact between the surface of the glass and the liquid phase leading to a reduction of the release rate of silver ions as can be shown in the evolution of the concentration of silver from 1 day of immersion in the fluid (Figure 5f).

However, the SEM micrographs of the as-prepared ABG4 glass upon immersion in SBF for 2 weeks (Figure 7) shown some cauliflower facets characteristic of hydroxyapatite (HA). Then, we may conclude that the most intense diffraction peak of AgCl at 14.69° overlaps with the main peak of HA masking the presence of calcium phosphate crystalline phases.

This issue was partially solved by incorporation of chitosan as a silver chelating agent. It is well established that chitosan as a polysaccharide biopolymer exhibits excellent chelating properties with metals and semiconductors due to the presence of both amino and hydroxyl groups in its monomers [39]. Then, ABG4 glass was embedded in 2 wt.% chitosan [39] and subsequently immersed in SBF for different time spans. The XRD diffraction patterns of the ABG4-BG composite before and after immersion in SBF (for 1D, 14D, and 21D) are shown in Figure 8. Before immersion in SBF (0D), a weak reflection of Ag was observed at 17.24° [40], which disappeared over the first hours. For all immersions lasting longer than 1D, the diffraction patterns showed reflections at 14.40°, 14.57°, and 14.77° together with intensive amorphous background which can be assigned to the characteristics reflections of hydroxyapatite (HA) [37]. Accordingly, incorporation of chitosan upon SBF testing of silver modified bioactive glass was shown to be an appropriate approach to reaffirm HA crystals formation.

### 3.3. Bactericidal Activity of the As-Prepared of the Silver Modified Glasses

To evaluate the bactericidal properties of the silver modified glasses, two different experiments were performed: Inhibition of bacterial growth and bactericidal activity. Figure 9a shows the inhibition of bacterial growth in the presence of the silver-containing as well as silver-free bioactive samples. It can be clearly observed that the silver-modified glassy materials (ABG3 and ABG4) inhibited bacterial growth, as revealed by the presence of four clean areas in the respective plates, while the silver-free BG sample shows a uniform bacterial growth. The bactericidal activity of the silver-modified samples over time is depicted in Figure 9b. Samples treated with silver-modified glasses showed a reduction of almost two orders of magnitude in bacterial number after one hour as compared to the silver-free BG sample (*p* < 0.05). Moreover, the number of bacteria treated with silver containing glasses decreased continuously up to reaching the detection limit after ca. 1.5 h and ca. 2 h for ABG4 and ABG3, respectively. Finally, samples treated with silver-free BG showed a slight decrease of bacterial number after one hour compared with the inoculum. However, after two hours of incubation at 37 °C, the number of bacteria increased to be comparable to the one of the inoculum. After four hours, the number of bacteria was one order of magnitude higher than that of the inoculum, indicating that BG has no bactericidal properties. The decrease of the number of bacteria observed in the silver free sample during the first hours of the analysis may be due to the release of F^−^ ions [41,42]. In conclusion, the silver-modified ABG glassy materials showed strong bactericidal activity against *E. coli*, which was not observed in the case of the silver-free parent BG.

## 4. Conclusions

The present study demonstrates that silver nanoparticles with sizes lower than 10 nm may be incorporated into the surface of BG glass using a facile, fast, and low-temperature synthesis route. According to the mid infrared spectra, the structure of ABG materials is similar to that of the parent glass BG, which indicates that the treatment performed within this work does not significantly alter the structure of the glass network, and thus does not interfere with its bioactivity process mechanisms.

Antibacterial tests showed that the silver-containing glasses, unlike the parent BG, inhibit the growth of *E. coli.* and exhibit rapid decrease in its viability, reaching the limit of detection after a maximum of 2 h.

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
