# Peer review of "Synthesis of Silver Modified Bioactive Glassy Materials with Antibacterial Properties via Facile and Low-Temperature Route"

_materials, 2020, doi:10.3390/ma13225115_

Round 1

Reviewer 1 Report

The paper reports the preparation and characterization of bioactive glass particles with surface decorated by silver nanoparticles. Bacterial growth tests demonstrated as the silver functionalisation leads to antibacterial activity against E. coli.

Even if the concepts presented in the study are not new and reported by many other papers, the authors reported a good and detailed characterization with many different technique trying to simplify the synthetic procedure with respect to other protocols (e.g. low temperature is used here).

I suggest the publication of the paper after the following revisions:

  • In the introduction, clarify the possible use of bioactive glasses in tissue engineering. What tissues are the target of these materials (I suppose bones)? What are the advantages of bioglasses with respect to other materials proposed for similar application? This introduction should make easier the article reading for a broad readership.
  • Related to point 1: Why do you study the mineralization of the materials? Moreover, Figure 6b has to be introduced/commented before Figure 7 (or Figure 6 has to be divided in 2 Figures following the text description).
  • Change the label of Figure 3 and 4.
  • The authors prepared materials with different amount of silver nanoparticles by changing the starting concentration of the silver salt. Can the author quantify the amount (or a relative weight ratio) of Ag NPs with respect to the bioactive glasses? How this parameter can affect the biological properties? Do materials with less amount of Ag NPs (ABG1 and ABG2) still preserve the antibacterial properties?

Author Response

Dear Reviewer 1,

thank you for the valuable and positive comments regarding our submission which we truly appreciate. We modified the manuscript according to your suggestions. You will find the changes made highlighted in yellow marking within the new version of the manuscript. The answers to your suggestions and comments outlined below in detail.

  1. The paper reports the preparation and characterization of bioactive glass particles with surface decorated by silver nanoparticles. Bacterial growth tests demonstrated as the silver functionalisation leads to antibacterial activity against E. coli.

Even if the concepts presented in the study are not new and reported by many other papers, the authors reported a good and detailed characterization with many different techniques trying to simplify the synthetic procedure with respect to other protocols (e.g. low temperature is used here).

Answer: Thank you for the positive feedback which we truly appreciate.

  1. In the introduction, clarify the possible use of bioactive glasses in tissue engineering. What tissues are the target of these materials (I suppose bones)? What are the advantages of bioglasses with respect to other materials proposed for similar application? This introduction should make easier the article reading for a broad readership.

Answer: Thank you for the comment. We agree with it and therefore we added a respective paragraph at the beginning of the introduction (page 1, lines 34-36).

  1. Related to point 1: Why do you study the mineralization of the materials?

Answer: The silver modified bioactive glasses investigated in the present work are intended for bone repair. The ability of the bioactive glass used in this work to enhance bone formation and to chemically bond to the surrounding bone tissue has already demonstrated (D.U. Tulyaganov, M.E. Makhkamov, A. Urazbaev, A. Goel, J.M.F. Ferreira, Synthesis, processing and characterization of a bioactive glass composition for bone regeneration, Ceramics International 39(3) (2013) 2519-2526). We decided that a comparison between the mineralization behavior of the bioglass and the bioglass modified with silver will be very interesting to identify whether the silver deposited on the surface of the materials may affect the intrinsic characteristics of the glass with respect to its bioactivity. According to our data showed in the manuscript, the mineralization properties of the glass are not significantly affected by the presence of silver on their surface.

  1. Moreover, Figure 6b has to be introduced/commented before Figure 7 (or Figure 6 has to be divided in 2 Figures following the text description).

Thank you very much for your input. Following your suggestion, Figure 6a and Figure 6b have been split into Figure 6 (page 9, lines 281-282) and Figure 8 (page 10, lines 301-303), respectively.

  1. Change the label of Figure 3 and 4.

We appreciate your remark and changed the respective captions of the figures in the manuscript (page 7, lines 241-242 and page 8, line 254).

  1. The authors prepared materials with different amount of silver nanoparticles by changing the starting concentration of the silver salt. Can the author quantify the amount (or a relative weight ratio) of Ag NPs with respect to the bioactive glasses?

Answer: Calculated relative weight ratios (further referred as Ag/BG) of Ag NPs with respect to the bioactive glass (e.g. considering that 1g of BG was suspended in 100 ml of AgNO3 aqueous solution) are as follows:

Sample

Ag/BG

ABG1

0.38/1.00

ABG2

0.83/1.00

ABG3

1.61/1.00

ABG4

2.37/1.00

  1. How this parameter can affect the biological properties?

Answer: The observation is indeed very interesting since an effect of Ag/BG on the biological properties is an important issue. According to the results of the present study the silver-modified ABG4 composition (having Ag/BG ratio 2.37/1.00) does not inhibit hydroxyapatite formation in vitro. Based on the data presented we cannot anticipate how this parameter can affect the biological properties. However, and evaluation would be very interesting, therefore we plan to perform in the near future in-vivo cell culture studies to fully assess how the ratio Ag/BG affect the biological properties.

  1. Do materials with less amount of Ag NPs (ABG1 and ABG2) still preserve the antibacterial properties?

Answer: In the present work we show the antibacterial behavior of the investigated materials increased by increasing the Ag/BG ratio. However, based on the data presented we cannot anticipate whether AGB1 and ABG2 have antibacterial properties. The investigation of the bactericidal properties of the silver modified glasses with less Ag/BG ratios is indeed, very interesting and therefore we will consider it in our forthcoming experimentation.

Thank you again for your effort and the helpful suggestions!

Yours sincerely,

Dr. Isabel Gonzalo de Juan

Reviewer 2 Report

1)Authors must be carefuly at the typing mistakes;

- for example at the row 151 and at the row 146 , the authors write 107 bacteria/mL; here correct is to write  107 CFU/mL (i.e. CFU  means  colony forming units)

- E. coli must be written with Italics characters

2) At materials and methods:

-authors must to specify in which way were made the determinations regarding CFU of E.coli

 - authors must specify clear in which way was estimated the antbacterial activity;

- from manuscript we understanding that the antimicrobial efect of glass with  or wihout Ag  against E. coli was made  in culture media type Luria Bertani which contain amplicilin.
The most strains of E coli are sensible to ampicillin. Or the bacterials developed from E.coli plasmids are not sensible to ampicillin?....Authors  must explain clear all these things or must to write the corect methodology used in determinations regarding antibacterial activity.

Author Response

Dear Reviewer 2,

thank you for the valuable and positive comments regarding our submission which we truly appreciate. We modified the manuscript according to your suggestions. You will find the changes made highlighted in blue marking within the new version of the manuscript. The answers to your suggestions and comments outlined below in detail.

  1. Authors must be carefuly at the typing mistakes; for example at the row 151 and at the row 146 , the authors write 107 bacteria/mL; here correct is to write 107 CFU/mL (i.e. CFU  means  colony forming units). E. coli must be written with Italics characters.

Answer: Thank you for your remark, we agree and therefore we removed bacteria/mL in the manuscript and we substituted it accordingly for CFU or CFU/mL. We also changed the typography of E. coli to italic format.

  1. At materials and methods: authors must to specify in which way were made the determinations regarding CFU of E.coli.

Answer: We agree and therefore added a more detailed paragraph in section 2.4.2, in which we speak about serial dilution and how to count colonies (page 4, lines 148-158).

  1. At materials and methods: authors must specify clear in which way was estimated the antbacterial activity;

Answer: The antibacterial activity analysis of the materials performed within this work were qualitative. We observed that in the areas treated with silver- modified glassy materials remained transparent, which indicated absent of bacteria growth. On the contrary, silver-free BG allowed bacterial growth in the areas that were treated with it, forming a uniform semitransparent-white layer through the whole plate. We agree that a more detailed explanation is needed and therefore added a more detailed paragraph in section 2.4.1. (page 4, lines 141-145).

  1. At materials and methods: from manuscript we understanding that the antimicrobial efect of glass with or wihout Ag  against E. coli was made  in culture media type Luria Bertani which contain amplicilin. The most strains of E coli are sensible to ampicillin. Or the bacterials developed from E.coli plasmids are not sensible to ampicillin?....Authors  must explain clear all these things or must to write the corect methodology used in determinations regarding antibacterial activity.

Answer: We used bacteria that are resistant to ampicillin, so ampicillin do not affect their growth. We added a more detailed paragraph at the beginning of section 2.4. to explain better this point (page 4, lines 129-135).

Thank you again for your effort and the helpful suggestions!

Yours sincerely,

Dr. Isabel Gonzalo de Juan

Reviewer 3 Report

This manuscript represents new, interesting and important improvement of the infection control in orthopaedical surgery. The issue is significant and the study is well designed and performed. Conclusions are based on the results.

However, there are reports that silver can change color of surrounding tissues, so in the discussion chapter should be few words on that.

Author Response

Dear Reviewer 3,

thank you for the valuable and positive comments regarding our submission which we truly appreciate. The answers to your suggestions and comments outlined below in detail.

  1. This manuscript represents new, interesting and important improvement of the infection control in orthopaedical surgery. The issue is significant and the study is well designed and performed. Conclusions are based on the results.

Answer: Thank you for the positive feedback which we truly appreciate.

  1. However, there are reports that silver can change color of surrounding tissues, so in the discussion chapter should be few words on that.

Answer: We agree that silver may change the color of the bioactive glass, for instance, in our study we observed that the white color of BG glass powder becomes grayish brown. However, considering the fact that the silver modified bioactive glass materials investigated are potential synthetic bone substitutes (i.e. will not be used for teeth repair) the colour is not a critical issue with respect to their application.

Thank you again for your effort and the helpful suggestions!

Yours sincerely,

Dr. Isabel Gonzalo de Juan

Round 2

Reviewer 1 Report

The authors reply successfully to the referee comments. Even if, I believe the study of biological properies with respect to the silver NPs content greatly increase the value of this article, the paper should be published in the present form. Maybe the authors should add some comments on this aspect (e.g. reporting in the paper some phrases of response 7 and 8 of the referee reports).